

# Sleep apnea detection from a single-lead ECG signal with automatic feature-extraction through a modified LeNet-5 convolutional neural network

Tao Wang[1], Changhua Lu[1,2], Guohao Shen[1] and Feng Hong[1]

[1] School of Computer and Information, Hefei University of Technology, Hefei, Anhui, China
[2] School of Software, Hefei University of Technology, Hefei, Anhui, China

## ABSTRACT

Sleep apnea (SA) is the most common respiratory sleep disorder, leading to some serious neurological and cardiovascular diseases if left untreated. The diagnosis of SA is traditionally made using Polysomnography (PSG). However, this method requires many electrodes and wires, as well as an expert to monitor the test. Several researchers have proposed instead using a single channel signal for SA diagnosis. Among these options, the ECG signal is one of the most physiologically relevant signals of SA occurrence, and one that can be easily recorded using a wearable device. However, existing ECG signal-based methods mainly use features (i.e. frequency domain, time domain, and other nonlinear features) acquired from ECG and its derived signals in order to construct the model. This requires researchers to have rich experience in ECG, which is not common. A convolutional neural network (CNN) is a kind of deep neural network that can automatically learn effective feature representation from training data and has been successfully applied in many fields. Meanwhile, most studies have not considered the impact of adjacent segments on SA detection. Therefore, in this study, we propose a modified LeNet-5 convolutional neural network with adjacent segments for SA detection. Our experimental results show that our proposed method is useful for SA detection, and achieves better or comparable results when compared with traditional machine learning methods.

Corresponding author
Tao Wang, wtustc@mail.ustc.edu.cn

# INTRODUCTION

Sleep apnea (SA) is the most common respiratory disorder, caused by partial or complete obstructions of the upper respiratory tract (*Li et al., 2018*; *Punjabi, 2008*). During sleep, SA events can occur hundreds of times, and, if repeated over a long period of time, can cause serious neurological and cardiovascular complications such as memory loss, high blood pressure, congestive heart failure, and poor cognitive ability during the day (*Khandoker, Palaniswami & Karmakar, 2009*; *Sharma & Sharma, 2016*; *Varon et al., 2015*; *Young et al., 1997*). Reportedly, approximately 5% of women and 14% of men have SA syndrome in the United States, and the incidence of the disease is increasing in various populations

(*Peppard et al., 2013*; *Song et al., 2016*). The severity of SA is clinically assessed using the apnea-hypopnea index (AHI). Subjects with an *AHI* > 5 combined with other symptoms (i.e., excessive sleepiness and poor cognitive ability during the day) are diagnosed with SA (*Marcus et al., 2012*; *Song et al., 2016*).

Polysomnography (PSG) is one of the most common tests used for SA diagnosis. It analyzes physiological signals (e.g., airflow, electroencephalogram (EEG), electrocardiogram (ECG), and respiratory signals) during sleep (*Bloch, 1997*; *Song et al., 2016*) in a hospital, and requires the patient to wear a number of electrodes and wires while an expert monitors the whole examination process. This complicated and uncomfortable examination experience has limited the application of PSG in clinical practice. To this end, several methods using a single channel signal (i.e., ECG (*Penzel et al., 2003*), SaQ$_2$ (*Hornero et al., 2007*), and respiratory sound (*Azarbarzin & Moussavi, 2013*)) for SA diagnosis have been proposed (*Song et al., 2016*) to reduce costs and to be more easily implemented. Among these, using an ECG signal has been the most popular method because it is one of the most physiologically relevant signals of SA occurrence and can be easily recorded using a wearable device.

For example, *Song et al. (2016)* developed a Hidden Markov Model (HMM) SA detection method using the frequency domain and time domain features extracted from EDR signals and ECG signals, and their model achieved an accuracy of 86.2% in per-segment SA detection. *Sharma & Sharma (2016)* proposed an RBF kernel LS-SVM for the per-segment SA detection based on features extracted from RR intervals by the hermit basic function, and the accuracy of their model was 83.8%. Existing methods mainly use frequency domain, time domain, and some nonlinear features acquired from ECG and its derived signals to construct the model. This requires researchers to have a wealth of relevant domain knowledge and experience, researchers with sufficient experience are uncommon. Recently, *Li et al. (2018)* proposed an SA detection method that uses stacked SAE to automatically extract features. Their method avoids over-reliance on ECG domain knowledge, and achieved an accuracy of 84.7% in per-segment classification. However, stacked SAE is essentially an unsupervised feature transformation that cannot extract features effectively (*Kang et al., 2017*).

A convolutional neural network (CNN) is a deep neural network that simulates the deep hierarchal structure of human vision (*Matsugu et al., 2003*). Compared to traditional machine learning methods, a CNN does not require hand-crafted features, and can automatically extract effective features through hierarchical layers. It has been successfully applied in speech recognition (*Abdel-Hamid et al., 2012*; *Palaz & Collobert, 2015*), image classification (*Sharif Razavian et al., 2014*; *Wei et al., 2016*), signal analysis (*Kwon, Shin & Kim, 2018*; *Sedighi et al., 2015*) and other fields. LeNet-5 is one CNN implementation with relatively few parameters and good performance (*El-Sawy, Hazem & Loey, 2016*; *LeCun, 2015*; *Wen et al., 2018*). It is worth noting that in a CNN, many parameters are prone to overfitting when training small data (i.e., the data used in this study), increasing the difficulty of this task. Therefore, the main objective of this study is to detect SA by automatically extracting features from RR intervals and amplitudes using LeNet-5. Previous studies (*De Chazal et al., 2000*; *Maier, Bauch & Dickhaus, 2000*; *Yadollahi & Moussavi,*
*2009*) have shown that adjacent segments offer useful information for SA detection. Additionally, we combine adjacent segments into our proposed method. Experimental results in the PhysioNet Apnea-ECG and UCD datasets show that our proposed method is robust, and its performance has been improved further since, promoting the clinical application of a single-lead ECG SA detection method.

# MATERIALS & METHODS

## Datasets

To ensure reliable results, two separate datasets were used in this study. A brief description of the two datasets is provided below.

### PhysioNet Apnea-ECG dataset

The first dataset was the PhysioNet Apnea-ECG dataset provided by Philipps University (*Goldberger et al., 2000*; *Penzel et al., 2000*). It contains a total of 70 single-lead ECG signal recordings (released set: 35 recordings, withheld set: 35 recordings), which were sampled at 100 Hz and ranged between 401 and 587 min. For each 1-minute ECG signal recording segment, the dataset provided an expert annotation (if there was an apnea event within this minute, it was labeled as SA; otherwise, normal). It was notable that there was no difference between hypopnea and apnea in the provided annotation file, and all events were either obstructive or mixed (central was not included). Additionally, these recordings were classified as Class A, Class B and Class C according to the Apnea–Hypopnea Index (AHI) value. Class A meant that the recording contained 10 or more SA segments per hour ($AHI \geq 10$) and the entire recording had at least 100 SA segments. Class B meant that the recording included five or more SA segments per hour ($AHI \geq 5$) and the entire recording contained five to 99 SA segments. Class C (or Normal) meant that the recording had less than five segments of SA per hour ($AHI < 5$).

### UCD dataset

The UCD dataset was the second dataset, which was collected by the University College Dublin, and can be downloaded from the PhysioNet website (https://physionet.org/physiobank/database/ucddb/). This dataset recorded the complete overnight PSG recordings of 25 (4 females and 21 males) suspected sleep disordered breathing patients, each contained 5.9 to 7.7 h of ECG signal as well as an annotation of the start time and the duration of every apnea/hypopnea event. Considering that this study primarily performed SA detection on 1-minute ECG signal segments, we converted continuous ECG data to 1-minute intervals which we correlated with annotations for normal and apnea events. According to the definition of apnea, an event should last at least 10s. However, an apnea event lasting 10s may be separated over two adjacent minutes, each having a smaller amount of apnea event time (*Mostafa, Morgado-Dias & Ravelo-García, 2018*; *Xie & Minn, 2012*). In the case of apnea or hypopnea lasting 5 or more consecutive seconds, the minute is considered to be an apnea. Additionally, each recording was classified as Class A, Class B or Class C by the Apnea–Hypopnea Index (AHI) value.

## Preprocessing

A method for automatically extracting features from RR intervals and amplitudes was developed in this study, and a preprocessing scheme was needed in order to obtain the RR intervals and amplitudes. Since several studies (*De Chazal et al., 2000*; *Maier, Bauch & Dickhaus, 2000*; *Yadollahi & Moussavi, 2009*) have shown that adjacent segment information is helpful for per-segment SA detection, the labeled segment and its surrounding $\pm 2$ segments of the ECG signal (five 1-minute segments in total) were all extracted for processing. We first used the Hamilton algorithm (*Hamilton, 2002*) to find the R-peaks, then used the position of the R-peaks to calculate the RR intervals (distance between R-peaks) and extract the values of the R-peaks (amplitudes). Considering that the extracted RR intervals had some physiologically uninterpretable points, the median filter proposed by *Chen, Zhang & Song (2015)* was employed. Since the obtained RR intervals and amplitudes were not equal time intervals, which was required by our proposed method, cubic interpolation was further employed, and 900 points of RR intervals and 900 points of amplitudes over 5-minute segments were obtained. The detailed preprocessing scheme is shown in Fig. 1.

## Convolutional neural network

In recent years, a CNN has been used as a research hotspot in the field of artificial intelligence (AI). It is a deep neural network method that simulates the deep hierarchal structure of human vision and has been successfully applied in image classification, natural language processing (NLP) and speech recognition (*Palaz & Collobert, 2015*; *Sharif Razavian et al., 2014*; *Yin et al., 2017*). Due to its proficiency in automatic feature extraction, CNN is also used to design advanced signal analysis methods (*Kwon, Shin & Kim, 2018*; *Sedighi et al., 2015*). For example, *Kiranyaz, Ince & Gabbouj (2015)* used CNN for ECG classification. Here, we used a simple and effective CNN implementation, LeNet-5, to construct our SA detection model. In the following section, we will introduce both the standard LeNet-5 and our modified LeNet-5.

### *Architecture of the standard LeNet-5*

The standard LeNet-5 proposed by *LeCun (2015)* was designed to solve the problem of character recognition. It consisted of an input layer, two convolution layers, two fully connected layers, two pooling layers and an output layer—in total, seven layers. The details of each layer are described in *LeCun (2015)*. Formally, a set of $N$ images $\{X_i, y_i\}_{i=1}^{N}$ are taken, where $X_i$ is the original image data and $y_i$ is a class category of the image (i.e., 0 and 1). The difference between the predicted label $\widehat{y_i}$ and the real label $y_i$ is calculated using the categorical cross entropy function, defined as follows:

$$J(\omega, b) \triangleq -\frac{1}{N} \sum_{l=1}^{N} y_{l1} \log \widehat{y_{l1}} + \cdots + y_{lK} \log \widehat{y_{lK}}$$

where $\omega$ and $b$ represent the weights and biases of the standard LeNet-5 network layers, respectively. $K$ is the number of class category and $\widehat{y_{lk}}$ corresponds to the softmax value of

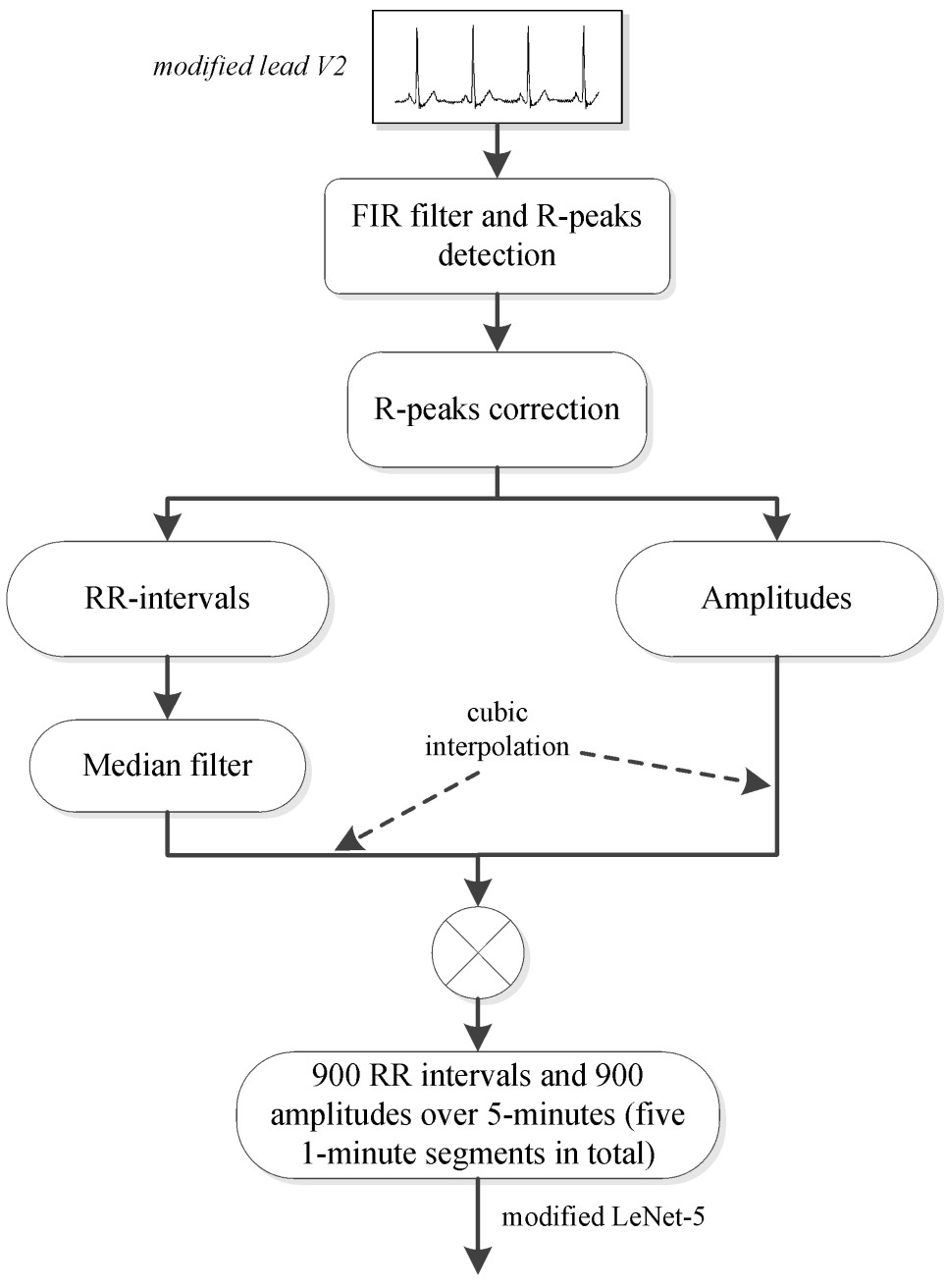

**Figure 1   PhysioNet Apnea-ECG dataset preprocessing scheme.** Note: In this study, the labeled segment and its surrounding ±2 segments of the ECG signal (five 1-minute segments in total) was extracted as a whole for processing.

the $k'th$ class category, defined as:

$$\widehat{y_{lk}} = \mathrm{softmax}(z_k) = \frac{e^{z_k}}{\sum_{i=1}^{K} e^{z_i}}$$

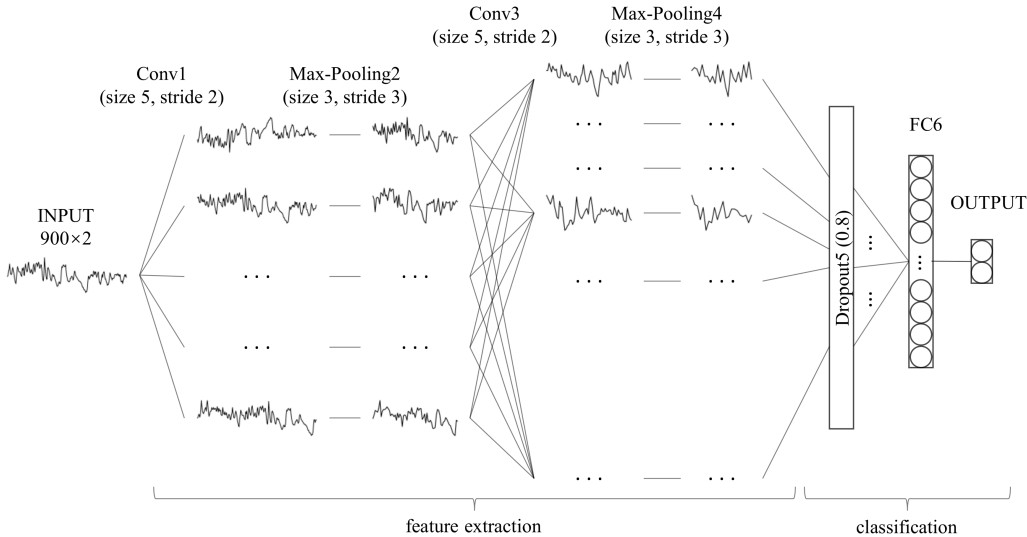

**Figure 2  Architecture of our modified LeNet-5.** It can be seen as a combination of convolutional neural networks (CNN) for feature extraction and full connection (FC, also known as MLP) as classifier.

where $z_i$ is the result of the corresponding $i'th$ class category of the last fully connected output. The weight and bias parameters of the convolutional operation and fully connected layers were learned by back-propagating (BP) the derivative of the loss with respect to parameters throughout the entire network (*Zeiler & Fergus, 2014*).

### Architecture of our modified LeNet-5

Unlike with character recognition, the time series used in this study had one-dimensional data, which is significantly different from two-dimensional character recognition problems. When compared with the millions of training samples in the field of character or image classification, the data samples used in this study were smaller, which increases the risk of overfitting. Moreover, SA detection is a binary classification problem that differs from character recognition. The feature maps, convolution layer strides and fully-connected layer nodes in the standard LeNet-5 may not be suitable for this scene. Therefore, we adjusted LeNet-5 as follows: (1) using a one-dimensional convolution operation instead of a two-dimensional convolution operation to feature extraction (*Kiranyaz, Ince & Gabbouj, 2015*); (2) adding a dropout layer between the convolution layer and fully connected layer to avoid over-fitting (*Srivastava et al., 2014*); (3) retaining only one fully connected layer to reduce network complexity (*Ma et al., 2018*); (4) modifying the size of the convolution layer strides and the number of fully-connected layer nodes. The architecture and details of our modified LeNet-5 are shown in Fig. 2 and Table 1, respectively. Compared to the standard LeNet-5, all convolution layer strides of our modified LeNet-5 were changed to two, and the number of feature maps was increased layer by layer. In particular, a dropout layer with a drop rate of 0.8 was added between the convolution layer and the fully connected layer, and the number of output layer nodes was reduced from 10 to two for our binary classification problem.

**Table 1  Details of our modified LeNet-5 convolutional neural network.**

| Layer | Parameter | Output Shape | Number[a] |
|---|---|---|---|
| Input | – | (None, 900, 2) | 0 |
| Conv1 | 32 × 5 × 2, stride 2, pad 0 | (None, 448, 32) | 352 |
| Max pooling2 | 3, stride 3, pad 0 | (None, 149, 32) | 0 |
| Conv3 | 64 × 5 × 2, stride 2, pad 0 | (None, 73, 64) | 10304 |
| Max pooling4 | 3, stride 3, pad 0 | (None, 24, 64) | 0 |
| Dropout5 | 0.8 rate | (None, 24, 64) | 0 |
| FC6 | 32, relu | (None, 32) | 49184 |
| Output | 2, softmax | (None, 2) | 66 |

**Notes.**
[a] The number of parameters generated by the corresponding operation.

## Experiment settings

In the field of SA detection based on a single-lead ECG signal, existing methods mainly extract suitable features based on expert experience, and then construct a model using the extracted features (*Sharma & Sharma, 2016*; *Song et al., 2016*; *Varon et al., 2015*), a process called feature engineering. In order to evaluate the performance of our proposed method, several popular feature engineering-based machine learning methods, including Support Vector Machine (SVM), K-Nearest Neighbor (KNN), Logistic Regression (LR) and Multi-Layer Perception (MLP), were employed for comparison. Various features that might have provided useful information for SA detection had been built in previous studies, and here we employed the features (RR intervals: 12 features, amplitudes: six features) that had an important effect on SA detection (*De Chazal et al., 2000*; *Song et al., 2016*) as the input of feature engineering-based methods. Table 2 lists the details of these features. Since some methods are sensitive to feature scales (i.e., KNN), the min-max normalization was used to normalize all features, which is defined as follows:

$$x^* = \frac{x - x_{\min}}{x - x_{\max}}$$

where $x$ is the feature to be normalized, and $x_{\max}$ and $x_{\min}$ are the maximum and minimum value in the features, respectively.

## Evaluation

By following (*Song et al., 2016*; *Varon et al., 2015*), the specificity (Sp), sensitivity (Sn), accuracy (Acc) and area under the curve (AUC) were employed to evaluate the performance of our proposed method, defined as follows:

$$\text{specificity} = \frac{TN}{TN + FP}$$

$$\text{sensitivity} = \frac{TP}{TP + FN}$$

$$\text{accuracy} = \frac{TP + TN}{TP + TN + FP + FN}$$

**Table 2  Feature set extracted based on previous studies.**

| Name | Derived from | | Details |
|------|------|------|---------|
| | RR[a] | Ampl[b] | |
| RMSSD | × | | Square root of the average of the squared difference between adjacent RR intervals. |
| SDNN | × | | Standard deviation of the difference between adjacent RR intervals. |
| NN50 | × | | Number of adjacent RR intervals exceeds 50 ms. |
| pNN50 | × | | NN50 divides by the number of RR intervals. |
| Mean RR | × | | Mean of RR intervals. |
| Mean HR | × | | Mean of heart rate (HR), which is derived from RR intervals. |
| Normalized VLF | × | × | Normalized very low frequency (VLF) component of the corresponding signal. |
| Normalized LF | × | × | Normalized Low frequency (LF) component of the corresponding signal. |
| Normalized HF | × | × | Normalized high frequency (HF) component of the corresponding signal. |
| LF/HF | × | × | The ratio of LF to HF of the corresponding signal. |
| LF/(LF + HF) | × | × | The ratio of LF to LF+HF of the corresponding signal. |
| HF/(LF + HF) | × | × | The ratio of HF to LF+HF of the corresponding signal. |

**Notes.**
[a] RR intervals of single-lead ECG signal.
[b] Amplitudes of single-lead ECG signal.

where *FP* and *TP* stand for "false positive" and "true positive", respectively. *FN* and *TN* represent "false negative" and "true negative", respectively.

## RESULTS & DISCUSSION

In this study, two separate datasets were used to validate our proposed method. The PhysioNet Apnea-ECG dataset was used as benchmark data to evaluate our proposed method's performance. The UCD dataset is an independent dataset we used to check the robustness of our proposed method against other datasets.

### Per-segment SA detection

Accurately predicting the presence of SA by given ECG segment (minute-by-minute) is key in this field, as it provides a solid foundation for the diagnosis of suspected SA patients. Therefore, we compared our proposed method with traditional machine learning methods on per-segment SA detection. The overall performance of the withhold set, including its specificity, sensitivity, accuracy and AUC, was used for comparison, as displayed in Table 3. As can be seen from Table 3, our modified LeNet-5 with automatic feature extraction performed well in all measurements with a specificity of 90.3%, sensitivity of 83.1%, accuracy of 87.6% and AUC of 0.950. Compared with the SVM that had the second highest accuracy, the overall performances were better by 6.0%, 6.2%, 6.2% and 0.063, respectively. It can also be seen from the results that KNN had the lowest prediction accuracy among the five methods, probably because the features extracted from the

**Table 3  The overall performance of our modified LeNet-5 and traditional machine learning methods in per-segment SA detection.**

| Method | Accuracy (%) | Sensitivity (%) | Specificity (%) | AUC |
|---|---|---|---|---|
| SVM | 81.4 | 76.9 | 84.3 | 0.887 |
| LR | 80.8 | 75.7 | 84.0 | 0.884 |
| KNN | 77.5 | 68.1 | 83.4 | 0.826 |
| MLP | 81.1 | 71.3 | 87.2 | 0.898 |
| LeNet-5 | 87.6 | 83.1 | 90.3 | 0.950 |

ECG signal were less spatially correlated and were not suitable for this scene, similar to the findings in literature (*Sharma & Sharma, 2016*). In summary, in per-segment SA detection, our proposed LeNet-5 with automatic feature extraction performed better than the commonly used feature engineering method.

## Per-recording classification

A recording consists of multiple one-minute ECG segments, and the classification of each recording refers to the overall SA diagnosis of these one-minute ECG segments, which is different from per-segment SA detection. Clinically, AHI is used to distinguish SA from normal recordings. Specifically, if the recording AHI is greater than 5, it is diagnosed as SA; otherwise it is considered to be normal. The recording AHI is calculated using the results of per-segment SA detection, which is defined as follows:

$$AHI = \frac{60}{T} * \text{num of OSA segments}$$

Where $T$ denotes the number of one-minute ECG segment signals, and $T/60$ is the hour of the entire recording. Therefore, AHI is employed here to diagnose the recording SA, and the above-mentioned measurement's accuracy, sensitivity, specificity and AUC are computed on the withheld set as listed in Table 4. It should be noted that the withheld set provided by the PhysioNet Apnea-ECG dataset had only 35 recordings, which may have resulted in low-precision per-segment methods showing better per-recording performance. By following previous studies (*Sharma & Sharma, 2016*; *Song et al., 2016*), the correlation value between the experimentally determined *AHI* and the actual *AHI* were also adopted to ensure the reliability of the comparison. As shown in Table 4, when compared with SVM, LR, KNN and MLP, our modified LeNet-5 with an accuracy of 97.1%, sensitivity of 100%, specificity of 91.7% and AUC of 0.996 performed better in per-recording classification. The correlation value of our modified LeNet-5 further confirmed this result, which increased by 0.091 when compared to the second highest SVM method.

## Effect of automatic feature extraction

In the previous parts, we discussed the overall performance of our proposed LeNet-5 in per-recording classification and per-segment detection. The results showed that, when compared with the existing methods, our proposed method significantly improved the performance in both per-recording classification and per-segment SA detection. Here, we will verify the power of the automatic feature extraction of our proposed method.

**Table 4** The overall performance of our modified LeNet-5 and traditional machine learning methods in per-recording classification.

| Method | Accuracy (%) | Sensitivity (%) | Specificity (%) | AUC | Corr.[a] |
|---|---|---|---|---|---|
| SVM | 88.6 | 100.0 | 66.7 | 0.978 | 0.852 |
| LR | 88.6 | 100.0 | 66.7 | 0.982 | 0.841 |
| KNN | 82.9 | 100.0 | 50.0 | 0.986 | 0.845 |
| MLP | 85.7 | 95.7 | 66.7 | 0.949 | 0.814 |
| LeNet-5 | 97.1 | 100.0 | 91.7 | 0.996 | 0.943 |

**Notes.**
[a] The correlation value between the actual *AHI* and the experimentally determined *AHI*.

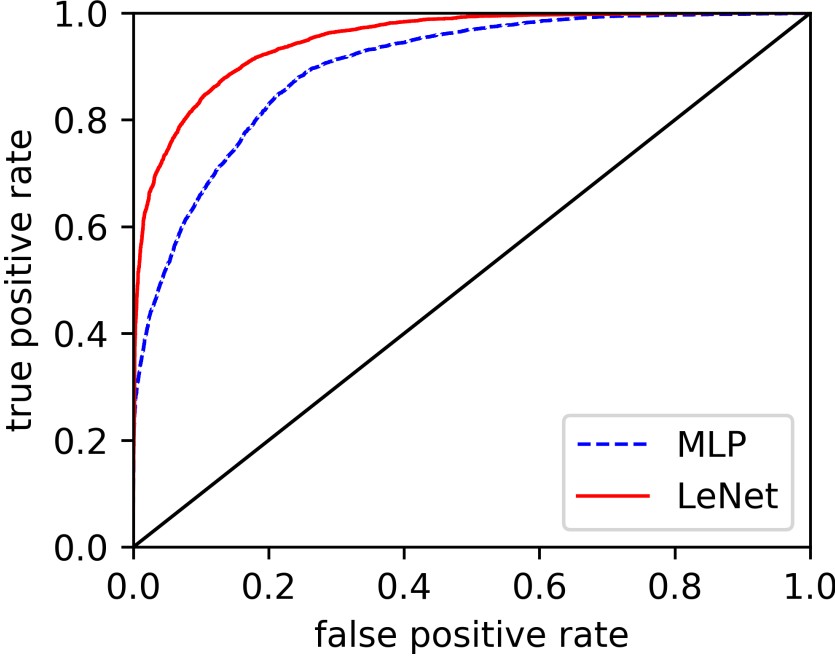

**Figure 3** Comparison of ROC curves of our modified LeNet-5 and MLP in per-segment SA detection.

Figure 3 displays the receiver operating characteristic (ROC) curves of our modified LeNet-5 and MLP in per-segment SA detection, since our modified LeNet-5 can be seen as a combination of convolutional neural networks (CNN) for feature extraction and full connection (FC, also known as MLP) as classifier (*Bae et al., 1998*; *Ludermir, Yamazaki & Zanchettin, 2006*), meaning that the effects of the automatic extraction features obtained by our proposed LeNet-5 and the features extracted by traditional feature engineering can be directly compared. As shown in Fig. 3, the LeNet-5's ROC curve is always above the MLP's ROC curve. These results suggest that the effect of the features extracted by our proposed automatic feature extraction method easily exceeded traditional feature engineering. Additionally, the measurements of the two methods in Table 3 also verify this result.

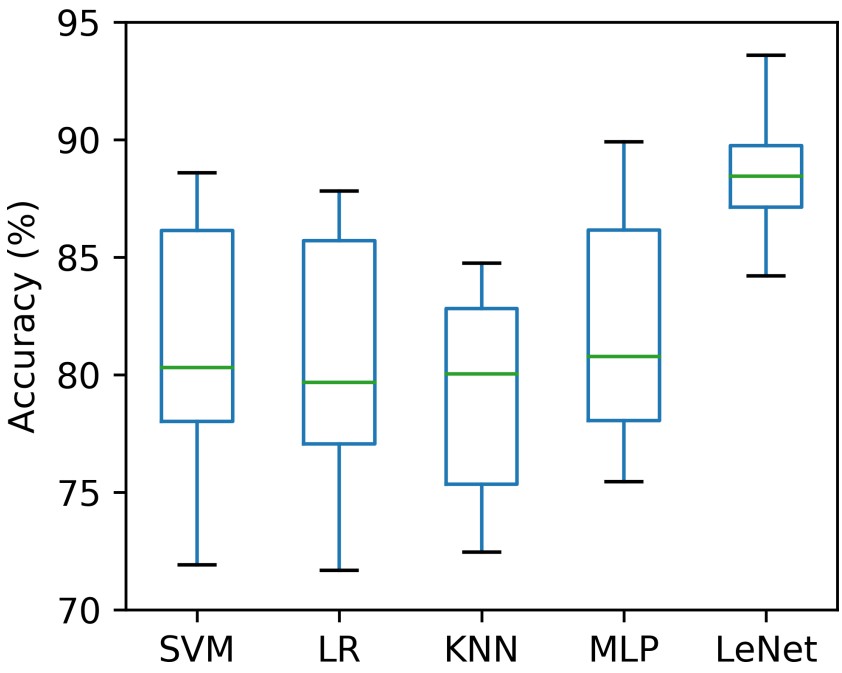

**Figure 4** Comparison of the per-segment detection accuracy of five classifiers calculated on 10 different test groups.

## Robustness Evaluation
### Ten-fold cross-validation

Validating a method with a single small-size test dataset may be biased or lead to incorrect results (*Sharma & Sharma, 2016*; *Song et al., 2016*). To this end, we used ten-fold cross-validation to ensure that our proposed method was robust under different test datasets. The whole dataset (70 recordings) was randomly split into 10 groups, of which nine were adopted to train the classifiers (SVM, LR, KNN, MLP and LeNet-5), and the remaining one was used for the test, taken 10 times. The accuracy of the per-segment SA detection calculated on 10 different test groups was drawn and is shown in Fig. 4. As seen in Fig. 4, the accuracies obtained using the SVM, LR, KNN, MLP and LeNet-5 ranged from 71.9% to 88.6% (mean ± standard deviation, 81.1% ±5.50%), 71.7% to 87.8% (mean ± standard deviation, 80.6% ± 5.47%), 72.5% to 84.8% (mean ± standard deviation, 79.3% ± 4.53%), 75.4% to 89.9% (mean ± standard deviation, 81.9% ± 4.98%) and 84.2% to 93.7% (mean ±standard deviation, 88.7% ± 3.05%), respectively. These results suggest that our proposed LeNet-5 with automatic feature extraction was more robust, and could achieve consistent and significantly better performances in different test datasets.

### Validation on UCD database

To ensure that our proposed method was robust in other datasets, we tested the performance of our modified LeNet-5 on an independent UCD dataset. Similar to the PhysioNet Apnea-ECG dataset, the dataset was divided into two parts, one for training and the other for verification. It is noteworthy that the original UCD dataset is continuously annotated based

**Table 5** The per-segment SA detection and per-recording classification performance in the UCDDB database.

| Classifier | Per-segment | | | Per-recording | | | |
|---|---|---|---|---|---|---|---|
| | Accuracy (%) | Sensitivity (%) | Specificity (%) | Accuracy (%) | Sensitivity (%) | Specificity (%) | Corr. |
| SVM | 70.6 | 32.7 | 83.3 | 92.3 | 100.0 | 50.0 | 0.251 |
| LR | 69.6 | 34.7 | 81.3 | 84.6 | 90.9 | 50.0 | 0.107 |
| KNN | 66.1 | 38.1 | 75.4 | 84.6 | 100.0 | 0.0 | 0.373 |
| MLP | 67.2 | 38.5 | 76.8 | 92.3 | 100.0 | 50.0 | 0.263 |
| LeNet-5 | 71.8 | 26.6 | 86.9 | 92.3 | 90.9 | 100.0 | 0.624 |

on the occurrence of events, which is different from the PhysioNet Apnea-ECG dataset, and we followed (*Mostafa, Morgado-Dias & Ravelo-García, 2018*; *Xie & Minn, 2012*) in converting them to 1-minute interval annotations. Table 5 shows the performance of our modified LeNet-5 and traditional machine learning methods in per-segment SA detection and per-recording classification. As shown in Table 5, the overall performance of different methods on the UCD dataset was worse than that of the PhysioNet dataset, caused by the small number of SA annotations on the UCD dataset. However, our modified LeNet-5 still had better or comparable performance to the traditional machine learning methods. For example, when compared with SVM in per-segment SA detection, the accuracy of our modified LeNet-5 was 1.2% better. In per-recording classification, our modified LeNet-5 had the same accuracy as SVM, but the correlation increased by 0.373. In general, our modified LeNet-5 is useful for SA detection.

## Comparison with existing works

So far, several works on SA detection based on a single-lead ECG signal have been published in the literature, and these works are mainly focused on feature engineering. Here, we compared our proposed method with relevant work that used both withheld sets and released sets of the PhysioNet Apnea-ECG dataset. However, a direct comparison was not available, due to the different samples sizes (*Li et al., 2018*). Table 6 shows the relevant work and performance of using the same dataset for per-segment detection. The released set was used for training, and the withheld set was used for validation. As shown, the classification accuracy of existing works ranged from 83.4% to 86.2%, which is lower than our proposed method (with an accuracy of 87.6%). It should be noted that *Li et al. (2018)* obtained the best sensitivity since their work is based on sensitivity optimization, while other works have focused on optimizing total classification accuracy. Table 7 lists the relevant pre-recording classification work and performance in which the same dataset is employed. It is noteworthy that, as we mentioned above, using traditional measurements to evaluate performance is not very accurate due to relatively small sample size (only 35 recordings in the withheld dataset), and the best method is to take the correlation value between the experimentally determined *AHI* and the actual *AHI* together, but not all works provide the correlation value. Nonetheless, our proposed method, with an accuracy of 97.1%, provides better or comparable performance than these works presented in the literature.

**Table 6** Comparison between the per-segment SA detection performance of our modified LeNet-5 and existing works.

| Reference | Features | Classifier | Accuracy (%) | Sensitivity (%) | Specificity (%) |
|---|---|---|---|---|---|
| *Varon et al. (2015)* | Feature Engineering | LS-SVM | 84.7 | 84.7 | 84.7 |
| *Song et al. (2016)* | Feature Engineering | HMM-SVM | 86.2 | 82.6 | 88.4 |
| *Sharma & Sharma (2016)* | Feature Engineering | LS-SVM | 83.4 | 79.5 | 88.4 |
| *Li et al. (2018)* | Auto encoder | Decision fusion | 83.8 | 88.9 | 88.4 |
| Our study | CNN | LeNet-5 | 87.6 | 83.1 | 90.3 |

**Table 7** Comparison between the per-recording classification performance of our modified LeNet-5 and existing works.

| Reference | Classifier | Accuracy (%) | Sensitivity (%) | Specificity (%) | Corr. |
|---|---|---|---|---|---|
| *Morillo & Gross (2013)* | PNN | 93.8 | 92.4 | 95.9 | – |
| *Sharma & Sharma (2016)* | LS-SVM | 97.1 | 95.8 | 100 | 0.841 |
| *Song et al. (2016)* | HMM-SVM | 97.1 | 95.8 | 100 | 0.860 |
| *Alvarez et al. (2010)* | LR | 89.7 | 92.0 | 85.4 | – |
| *Li et al. (2018)* | Decision fusion | 100 | 100 | 100 | – |
| Our study | LeNe-5t | 97.1 | 100.0 | 91.7 | 0.943 |

## CONCLUSIONS

In this study, we developed an SA detection method based on modified LeNet-5 and adjacent ECG segments. Experimental results showed that our proposed method is useful for SA detection, and the performance of our method is better than both traditional machine learning methods and existing works. Due to the high precision requirements of clinical applications, further improvements in our proposed method will accelerate the development of ECG-based SA detection devices in clinical practice. Furthermore, since only a single-lead ECG signal is used, our proposed method can also be used to develop SA detection for home healthcare services using wearable devices. However, our proposed method has some limitations. Because the Apnea-ECG dataset is labeled in 1-minute segments, an apnea/hypopnea event could occur in the middle of two 1-minute segments and a 1-minute segment could contain more than one apnea/hypopnea event. Additionally, the dataset does not separately label hypopnea and apnea events in the provided annotation file, and all events are either obstructive or mixed (central is not included). This could mean our proposed method cannot distinguish between hypopnea and apnea, and cannot detect central events. In future research, we will include other datasets to solve the above problems.

## ACKNOWLEDGEMENTS

The authors would like to thank Dr. Thomas for providing the data.

### Funding

The work is supported by the Special-funded Program on National Key Scientific Instruments and Equipment Development of China under grant No. 2013YQ220643, and the National Natural Science Foundation of China under grant No. 71331002. The funders had no role in study design, data collection and analysis, decision to publish, or preparation of the manuscript.

### Grant Disclosures

The following grant information was disclosed by the authors:
Special-funded Program on National Key Scientific Instruments and Equipment Development of China: 2013YQ220643.
National Natural Science Foundation of China: 71331002.

### Competing Interests

The authors declare there are no competing interests.

### Author Contributions

- Tao Wang conceived and designed the experiments, performed the experiments, analyzed the data, contributed reagents/materials/analysis tools, prepared figures and/or tables, authored or reviewed drafts of the paper, approved the final draft.
- Changhua Lu conceived and designed the experiments, authored or reviewed drafts of the paper, approved the final draft.
- Guohao Shen and Feng Hong conceived and designed the experiments, contributed reagents/materials/analysis tools, approved the final draft.

### Data Availability

The codes are available in a Supplemental File.

The PhysioNet Apnea-ECG dataset is available at http://www.physionet.org/physiobank/database/apnea-ecg/ (DOI:10.13026/C23W2R).

Notes: In Linux environment, you can download the dataset directly using the following commands: "wget -np -r http://www.physionet.org/physiobank/database/apnea-ecg/".

### Supplemental Information

Supplemental information for this article can be found online at http://dx.doi.org/10.7717/peerj.7731#supplemental-information.

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
