# Peer review of "Sleep apnea detection from a single-lead ECG signal with automatic feature-extraction through a modified LeNet-5 convolutional neural network"

_PeerJ, doi:10.7717/peerj.7731_

## Round 0.1 · original submission · Major Revisions

Your work has undergone 2 independent reviews that recommended major revisions.

Primarily, your conclusions are not valid given that you have not compared to other existing published works. Please comment on the application of your work to detect different types of sleep disturbances. Validation of your algorithm with another publicly available database (University College Dublin) will add strength to your work.

[]

Reviewer 1 ·

Basic reporting

In their manuscript “Obstructive sleep apnea detection from single-lead 1 ECG signal with automatic feature-extraction through improved LeNet-5 convolutional neural network“, the authors used a standardized dataset (PhysioNet Apnea-ECG dataset) and applied convolutional neural network, a component of artificial intelligence which can automatically learn the underlying rules and features of objects.
- Wording and grammar need to be improved. I propose to get this manuscript proof read by a native speaker.
- If I understand the manuscript correctly, the authors were the first to use CNN in the field of sleep apnea detections. This could be further clarified in the manuscript.

Experimental design

- This is not a specialized artificial intelligence journal. Please explain further how exactly CNN works.
- The authors mention, that OSA episodes were detected. Were just “obstructive” sleep apnea episodes or also “central” and “mixed” apnea episodes detected. I would propose to avoid the use of “OSA” and to use “sleep apnea episodes” instead. Please provide further PSG data: Obstructive vs central apneas. Sleep apnea episode durations…

Validity of the findings

Methodological comments:
- The method is not tested prospectively in an additional dataset, which limits the novelty and the clinical implications.

Statistical comments:
- One limitation is the small dataset to validate/ develop the algorithm.
- Are there statistical/ methodological rules to assess the minimal size of the validation cohort?

Additional comments

The manuscript is interesting and of potential clinical importance. However, there are several points (including revision of wording and grammar) which should be considered in a revised version of the manuscript. The introduction is too long and should be shortened to better focus on the hypothesis and goal of the study.

·

Basic reporting

-The article is written in professional English. However, it has some gramatical mistakes that should be amended and/or revised.

-The article includes sufficient background with references. However, the advantages of deep learning (CNN) over traditional approaches could be better explained.

-The article has an appropriate structure, including figures and tables. Raw data is also indicated and code is available. The only suggestion is that the different lines in the code files could contain more text comments in such a way that the readers can understand it better.

Experimental design

-Authors indicate how the research fills an identified knowledge gap. In this sense, they indicate at the end of the introduction that they propose an improved Le-Net with adjacent ECG segments for OSA detection. However, I would indicate explicitly the hypothesis and the goal of the study, since it could enhance the manuscript's readability.

-Methods are in general well described. Nonetheless, I would add a figure/scheme that illustrates the preprocessing stage (lines 110-122).

Validity of the findings

-Data employed in this study come from a well-known database (Apnea-ECG). However, this database has some limitations:
i). There is no differentiation between apneas and hypopneas.
ii).Annotations are made for each 1-min segment. However,in most of the databases annotation files for apneas/hypopneas contain the start and end of the events. Thus, a 1-min segment could contain more than one apnea/hypopnea event, or an apnea/hypopnea event could occur in two different 1-min segments. Thus, the approach of labeling each 1-min segment as apnea or non apnea may not be applicable to other databases. Nonetheless, authors indicate as a future goal the necessity of using another datasets.

-The conclusion stated in the abstract is that their method achieves similar or better resuts in comparison with existing works in the literature. The conclusion is derived from the results. However, I think that this conclusion could be better stated. The conclusion of the study should be that, based on the results obtained, their proposal (a deep learning approach based on a modified LeNet-5 with adjacent ECG segments) is useful for OSA detection.

Additional comments

-Lines 55-57. With respect to the following text“Among them, the ECG signal is the most concerned signal because it can show the physiological characteristics of OSA occurrence and facilitates recording using a wearable device.” Other signals can also be recorded using a wearable device (pulse rate and blood oxygen saturation). Authors should justify better the use of ECG instead of other signals from polysomnography.

-Authors should indicate the advantages of deep learning methods (such as CNN) over traditional approaches around line 76.

-Authors state that use an improved Le-Net in the title and introduction. I think that it would be more correct to say that a modified version of Le-Net is used.

-Line 91. Authors indicate at the end of the introduction that they propose an improved Le-Net with adjacent ECG segments for OSA detection. However, I would indicate in a more explicit way which is the hypothesis and the goal of the study.

-Lines 98-109. Authors should indicate that an OSA minute means that there is an apnea or hypopnea in this minute. This would enhance readability.

-Lines 110-122. Authors should include a figure/scheme in order to better explain this process. This would help to understand it. Do authors mean distance between R peaks when they say RR intervals? Do authors mean values of the R peaks when they say RR amplitudes? In the line 112, authors could put an explanation such as “ (five 1-min segments in total) ” after “the labeled segment and its surrounding ±2 segments of the ECG signal”.

-Lines 124-132. Authors should indicate which properties of a signal CNN can measure in such a way that CNN are useful for signal analysis.

-Lines 157-161. Authors should use references to support statements 1) to 4).

-Lines 183-190. The use of positive and negative predictive values and the use of positive and negative likelihood ratios could also be useful to assess the performance of your proposal.

-It should be more clear that 5 minutes of the ECG signal are used to predict the presence of apneas and hypopneas in each 1-min segment.

-Authors should analyze the reasons why their CNN approach may have obtained better results than the machine learning classifiers the as well as the results reported by previous studies. That is to say, which properties of the CNN and the RR features have made their method obtain better results.

-Line 268. I think that it should be ”Table 5” instead of ”Table 3”.

-There are some problems of the Apnea-ECG database that could affect when using the CNN trained in another databases: i) an apnea/hypopnea event could occur in the middle of two 1-min segments; ii) a 1-min segment could contain more than an apnea/hypopnea event. These are limitations of the study.

-Authors say that their method should be applied to other datasets as a future goal. In this sense, there is another public database the St. Vincent's University Hospital / University College Dublin Sleep Apnea Database (https://physionet.org/pn3/ucddb/) that contains ECG recordings. In the ucddb database, apnea/hypopnea events are indicated in the annotation text files by the start and duration of the event. In this respect, it would be really interesting to see the performance of the CNN model to detect apneas, hypopneas and estimate the AHI of the patients of the ucddb database.

- The conclusion of the study should be that, based on the results obtained, their proposal (a deep learning approach based on a modified LeNet-5 with adjacent ECG segments) is useful for OSA detection.

---

## Round 0.2 · Major Revisions

Your revised manuscript has undergone further peer review.
The outstanding concerns of the reviewer need to be addressed.
Further the standard of the English Language remains inadequate.
The writing needs major revision:

- Language and grammar
- Introduction is still too long
- Some results are somehow shown in discussions
- The clinical implications are not conveyed
- The conclusions is too lengthy
- References in conclusions should be avoided
- please specify how the small increase in sensitivity/specificity/accuracy will change clinical practice
- When validated against the UCDDB, the performance is much worst.

This indicates that your method along with others require further work. Please comment and discuss in the manuscript

·

Basic reporting

-English language has been improved. There are still some grammatical mistakes but not as many as in the original version of the manuscript.

Experimental design

-Authors indicate their proposal in the introduction of the study. However, they should indicate explicitly the hypothesis and the goal of the study (we hypothesize that, the hypothesis of this research is, our goal is, the main objective of the study is ….), since it could enhance the manuscript's readability.

-Methods are in general well described. Nonetheless, in methods and results sections it seems that authors are using only Apnea-ECG database, whereas in the discussion they also comment the validation on the UCDDB database. Authors only comment the use of the UCDDB database in the discussion section. I think that this study uses two public databases: Apnea-ECG an UCDDB. Thus, the use of both datasets should be indicated in the materials and methods section of the manuscript. In addition, results of the UCDDB recordings should also be included in the results section of the manuscript. This would really give to the readers the idea that this research has employed both datasets.

Validity of the findings

-Authors have proven the validity of their methodology using two different databases: Apnea-ECG database and UCDDB database.

Additional comments

-Despite authors effort to address my comment, the hypothesis and the goal of the study care not explicitly indicated in the manuscript. In this sense, they should indicate explicitly the hypothesis and the goal of the study (we hypothesize that, the hypothesis of this research is, our goal is, the main objective of the study is ….), since it could enhance the manuscript's readability.

-Lines 113-114: “Therefore, the labeled segment and its surrounding ±2 segments of the ECG signal (five 1-minute segments in total) are extracted as a whole for further processing” I would include this explanation in the figure 1 that authors have included in the manuscript following my previous review.


-Authors only comment the use of the UCDDB database in the discussion section. I think that this study uses two public databases: Apnea-ECG an UCDDB. Thus, the use of both datasets should be indicated in the materials and methods section of the manuscript. In addition, results of the UCDDB recordings should also be included in the results section of the manuscript. This would really give to the readers the idea that this research has employed both datasets.

---

## Round 0.3 · accepted · Accept

Thank you for all the improvements made.